# Optimizing the in vitro production of immunomodulatory cells for the induction of tolerance in solid organ transplantation

**Nils Ågren**[ID]*, **Ming Yao, Carl Skantze, Keyvan Habibi, Bo-Göran Ericzon**[ID], **Makiko Kumagai-Braesch**

Intervention and Technology (CLINTEC), Transplantation Surgery, Karolinska Institute, Stockholm, Sweden

* nils.agren@ki.se

## Abstract

### Background

Cell therapy can be utilized to induce operational tolerance following solid organ transplantation. In thi study, donor-specific immunomodulatory cells (DSIMC) are generated by co-culturing recipient peripheral blood mononuclear cells (PBMC) with irradiated donor PBMC in the presence of belatacept, a CTLA-4-IgG1 fusion protein. DSIMC promote a regulatory response to donor cells. Reinfusion of these cells into the recipient may induce donor-specific tolerance, enabling weaning or complete cessation of immunosuppression (IS).

### Aim

This study aims to determine optimal culture conditions for DSIMC production.

### Methods

DSIMC were generated by culturing PBMCs from healthy volunteers with irradiated allogeneic PBMC and belatacept. We evaluated the choice of medium, plasma supplementation, costimulation blocker concentration, and red blood cell (RBC) lysis, using automated cell counts, cytokine assays, PCR, and flow cytometry.

### Results

Increasing belatacept concentration (0-80 µg per million cells) resulted in a significant reduction in PD-1 expression in regulatory T cells. RBC lysis reduced inflammatory cytokine production and improved DSIMC generation, as indicated by increased IL-10 and decreased IFN-γ production. Between culture days 9-14, the total cell yield and IFN-γ-producing cell numbers declined, while IL-10-producing cell numbers increased.

**Citation:** Ågren N, Yao M, Skantze C, Habibi K, Ericzon B-G, Kumagai-Braesch M (2025) Optimizing the in vitro production of immunomodulatory cells for the induction of tolerance in solid organ transplantation. PLoS One 20(11): e0333356. https://doi.org/10.1371/journal.pone.0333356

**Data availability statement:** All relevant data are within the paper and its Supporting Information files.

**Funding:** This study was supported in part by research grants from the Swedish Research Council (Vetenskapsrådet, VR2018-00845), the regional agreement on medical specialization training and clinical research between the Stockholm County Council and Karolinska University Hospital (20180606), CIMED (Center for Innovative Medicine, FoUI-975210), and Gelinfonden. Bristol-Myers Squibb (BMS) provided belatacept for experimental use but had no role in the study design, data collection, analysis, or manuscript preparation.

**Competing interests:** The authors have declared that no competing interests exist.

## Conclusion

Treating responder and irradiated stimulator PBMC with RBC lysis buffer before co-culture in TexMACS medium supplemented with 1% autologous plasma and 40 µg belatacept per million cells for 14 days produces DSIMC which meet established quality criteria. The protocol is currently currently under evaluation in a clinical study.

## Background

Currently, immunosuppression (IS) is needed after solid organ transplantation (SOT) to prevent rejection [1]. By design, conventional IS induces immune system dysfunction [1], and aside from being associated with short-term acute side effects, is also associated with an increased risk of infection [2] and cancer [3]. IS drugs are also frequently diabetogenic [4,5], nephrotoxic [6,7], and associated with cardiovascular side effects [8,9]. Furthermore, the risk of rejection remains present, and lifelong IS therapy is thereby required to protect the transplanted organ [10], and despite best practices, chronic rejection may still lead to the loss of graft function. Due to the scarcity of organs and the need to reduce the need for retransplantation, novel therapeutic strategies are needed to improve the long-term outcomes after transplantation [10].

In SOT, operational tolerance is defined as a tolerogenic state of the recipient towards the donor organ without external administration of IS therapy, and which has lasted for over 1 year. Historically, it has been noted that some transplant patients, especially liver recipients, can achieve such tolerance either deliberately by IS weaning protocols or by lack of compliance [11,12]. The mechanism for the development of naturally occurring operational tolerance is not fully understood. The recipient immune system may either be made tolerant or unresponsive to antigens in the donor tissue [11], or the immune reaction may be tempered by other immune cells, such as regulatory T cells [12–15]. This has led to an interest in the idea of promoting this type of tolerance by other means [12].

One promising strategy for achieving tolerance, highlighted by the success of a Japanese pilot study, involves inhibiting immune activation through blockade of the costimulation complex [16,17]. This strategy seems to render peripheral T cells anergic, leading to a regulatory response toward donor tissues [16]. Recipient cells are co-cultured with irradiated donor cells and a costimulation blocker in a one-way mixed lymphocyte culture, generating donor-specific immunomodulatory cells (DSIMC). Generated cells can then be infused into the recipient after initial IS treatment to promote tolerance to donor antigens, likely through infectious tolerance [16,17]. These cells have been shown to have donor-specific suppressive ability but also general anti-inflammatory properties [18]. The strategy has thus been shown to be efficacious both *in vitro* [18] and *in vivo*, in primate as well as in clinical trials [16,17,19]. In these studies, immune cells from irradiated living donors (LD) were cultured together with recipient cells with monoclonal CD80 and CD86 antibodies to block the CD28-mediated costimulation complex [16,17,19]. In the clinical studies, these cells were then reintroduced to the organ recipient after two weeks in culture, and up to 70% of

patients remained free from IS after five years [20]. The demonstrated advantages include a reduced need for traditional IS while preserving pathogen-specific immunity and achieving improved control of long-term graft rejection.

Our research group is attempting to adapt this strategy to liver transplantation with brain-dead donors (BDD) in Sweden. However, to conform to Swedish regulations, the ex-vivo cell generation protocol has had to be modified. Mouse monoclonal CD80 and CD86 antibodies are not accepted by the Swedish medical products agency. Due to this, we instead use belatacept, a widely approved and established drug in transplantation, with a well-known safety profile [21]. Belatacept is a CTLA-4-Ig (Cytotoxic T-Lymphocyte Antigen 4 Immunoglobulin) fusion protein that binds to the same CD28-mediated costimulatory complex as CD80/86 antibodies and has previously been used *in vitro* with comparable results [22]. In a previous study we used cells from healthy blood donors to simulate the co-culture of organ donor and recipient cells as stimulators and responders, comparing the effects of belatacept and mouse anti-human CD80/CD86 antibodies, and found their effects comparable [23].

To establish a cell generation protocol suitable for clinical trials, we performed an analysis of culture condition variables. The culture medium was changed to TexMACS due to GMP (Good Manufacturing Practice) requirements, and the cell concentration needed to be defined for maximum viability and yield. In the clinical trial, organ recipient and donor peripheral blood mononuclear cells (PBMC) will be collected through leukapheresis. Because these PBMCs are inevitably contaminated with a substantial amount of red blood cells (RBC) which confound cell counting and may influence cell-cell interactions, we also evaluate the effects of RBC lysis. We initially chose the belatacept dose based on a study by Davies et al. [22]. However, the optimal dose of belatacept in our culture setting had not been determined.

The aim of this study is to ascertain optimal culture conditions to produce DSIMC through the co-culturing of recipient PBMC with irradiated PBMC from healthy control and belatacept and which achieve high immunomodulatory function with a high cell yield. For this purpose we investigated (1) different cell culture media, (2) the addition of 1% plasma to TexMACS cell culture medium, (3) the effect of red blood cell lysis, (4) the effect of belatacept concentration, (5) the development of generated cell immunomodulatory function over time, and (6) remaining stimulator cell numbers and the fate of irradiated stimulator cells.

## Methods

### Blood collection

The lymphocytes used in these experiments were extracted from buffy coats from healthy blood donors and supplied through a local blood bank (Blodcentralen, Karolinska University Hospital, Huddinge). Blood samples were collected between 30/11/2020 and 29/09/2022. Ethical approval regarding the use of human materials was given by the Swedish Ethical Review Authority (ref. nr. 2020–02075) for the use of blood products from healthy volunteers and BDD. This study follows the recommendations in the declaration of Helsinki regarding informed consent. Appropriate measures to protect the privacy of the blood donors were undertaken. The blood donors were informed regarding the possible use of their blood for research purposes and supplied their written consent to a legally authorized representative.

### Plasma preparation

The buffy coats were centrifuged at 400 × g for 10 minutes. The supernatant plasma layer was then collected and centrifuged at 2500 × g for 15 minutes at 4°C. The plasma was heat-inactivated for 30 minutes at 56°C, followed by a final centrifugation step at 2500 × g for 15 minutes at 4°C to remove the precipitate and collect the purified plasma. The plasma was used either on the day of preparation or kept frozen (−20°C) until needed.

### Lymphocyte separation

The PBMCs were extracted from the buffy coats through density gradient separation with the use of Lymphoprep and SepMate tubes (Stemcell Technologies, Vancouver, Canada) according to the protocol supplied by the manufacturer. The

cells were either used fresh or aliquoted and frozen in 10% DMSO (Merck Life Science AB, Solna, Sweden) for later use. The stimulator (simulated donor) PBMC were irradiated with 35 Gy with an x-ray irradiator (Raycell Mk2, Best™ Theratronics, Ottawa, Canada), and used fresh or aliquoted as described above.

### Cell culture conditions

Freshly separated recipient and donor PBMCs were co-cultured at a ratio of 5:2, with $50 \times 10^6$ responder cells and $20 \times 10^6$ irradiated stimulator cells in 15 mL TexMACS medium supplemented with 1% autologous plasma and belatacept (40 µg/$10^6$ cells, NULOJIX, Bristol-Myers Squibb, Princeton, New Jersey, USA). The cultures were maintained in 25 cm$^2$ flasks at 37°C with 5% $CO_2$. On day 7, the medium was refreshed, and cultures were replenished with $20 \times 10^6$ thawed irradiated stimulator cells and belatacept. ABO-compatible donor-recipient pairs were used throughout the study.

### Comparison of culture media

We compared the culture media AIM-V (serum-free medium with Streptomycin Sulfate and Gentamicin, Thermo Fisher Scientific, Waltham, Massachusetts, USA), ALyS505N-0 (Funakoshi, Tokyo, Japan), and TexMACS (Miltenyi Biotec, Bergisch Gladbach, Germany) regarding cell yield, viability and composition. AIM-V was previously used in a primate study [16] and ALyS505N-0 in a clinical pilot study [17] to generate immunomodulatory cells. In these experiments, all media were supplemented with 1% autologous responder plasma and belatacept (40 µg/$10^6$ cells).

### Effect of addition of autologous plasma

DSIMC were generated with and without the supplementation of 1% autologous plasma in conditions similar to previous culture experiments, using the same method as described above. The cultures were compared regarding cell yields, viability, and composition.

### Effect of RBC lysis

PBMC samples were washed with phosphate-buffered saline (PBS) three times and then treated with 5 mL ammonium chloride potassium (ACK) lysis buffer, (Thermo Fisher Scientific, Waltham, Massachusetts, USA) or kept in PBS for 10 minutes at room temperature and washed three times. DSIMC yield, viability, phenotype, and cytokine profiles were compared after 14 days.

### Kinetics of DSIMC development

After medium change and culture reconstitution with thawed stimulator PBMC and belatacept on day 7, cells were harvested after a further 2, 5, or 7 days, with a total culture time of 9, 12, and 14 days. The cell yields, viability, composition, and cytokine profiles of the generated cells were compared. In a subset of experiments, HLA-A2 (Human Leukocyte Antigen A2) serotype status was determined through FCM, and mismatched pairs were co-cultured as responder and stimulator cells. After 14 days of culture, the cells were stained with anti-HLA-A2 (*BB7.2*, BioLegend) and the origin was determined through FCM.

### Belatacept concentration

We evaluated the concentration of belatacept in our setting. The DSIMC were generated with belatacept concentrations of 0, 0.1, 1, 10, 40, and 80 µg/$10^6$ cells (0, 0.33, 3.3, 33, 133, and 266 µg/mL). Culture supernatant and generated cells were harvested on day 14, and cell yields, viability, composition, and cytokine production were compared.

## Cell counting and viability assay

Viability and cell counts were measured using Acridine Orange and DAPI (solution 13) with an automated cell counter (NC-3000, Chemometec, Allerød, Denmark). These counts were verified and compared with manual counting using Trypan Blue (Thermo Fisher Scientific) and an optical microscope (Nikon).

## Cell phenotyping

Cell phenotypes were analyzed before and after culture using flow cytometry (FCM, FACS Canto, BD Biosciences, BD, Franklin Lakes, New Jersey, USA). The following Lineage markers were used: monocytes (CD14, *M5E2*, BD Biosciences, BD, Franklin Lakes, New Jersey, USA), B cells (CD19, *HIB19,* BD Horizon, BD), T cells (CD4, *L200,* CD8, *SK1,* BD Pharmingen, BD), and NK cells (CD16, *3G8*, CD56, *B159*, BD Pharmingen). CD25 (*BC96*, BioLegend, San Diego, California, USA), CD127 (*hIL-7R-M21*, BD Pharmingen), and FoxP3 (*236A/E7*, BD Pharmingen) were used as Treg markers. PD-1 (*NAT105,* BD Biosciences, *Programmed Cell Death Protein 1*) was used as a T cell activation marker. For intracellular staining (FoxP3), PBMC were permeabilized using a permeabilization kit (*Cytofix/Cytoperm*, BD Horizon). Tregs were defined as CD4+CD25+CD127lowFoxP3+. The FCM gating strategy is shown in S1 Fig. The methods have previously been described and/or follow protocols supplied by manufacturers.

## Supernatant cytokine assay

Culture supernatant cytokine concentrations were measured using a multiplex electrochemiluminescence (ECL) immunoassay (MSD V-Plex Proinflammatory Panel 1 Human Kit, Meso Scale Diagnostics, Rockville, Maryland, USA). Cytokines measured included IFN-γ (interferon gamma), IL-1β (interleukin 1 beta), IL-6 (interleukin 6), IL-10 (interleukin 10), TNF-α (tumor necrosis factor alpha), and IL-17 (interleukin 17). The method followed the protocol supplied by the manufacturer.

## Antigen-specific cytokine assay

After co-culture harvest, the generated cells were restimulated with stimulator cells, and the production of IFN-γ and IL-10 was measured using an ELISpot (Enzyme-Linked ImmunoSpot) assay (Mabtech AB, Nacka, Sweden). The general method followed the protocol supplied by the manufacturer. In brief, generated DSIMC were seeded ($2x10^5$/well) and incubated with medium alone, with the addition of irradiated stimulator cells ($2x10^5$/well), or with monoclonal CD3 and CD28 antibodies (1 µg/mL) for 24 hours at 37°C in a humidified atmosphere with 5% $CO_2$. Biotinylated detection antibodies were added to each well and incubated for one hour. Plates were washed with PBS, and ALP-conjugated streptavidin was added. The plates were washed again with PBS, and the substrate (BCIP/NBT, 5-Bromo-4-Chloro-3'-Indolylphosphate/Nitro blue Tetrazolium Chloride) was added. The reaction was stopped by washing the plates with tap water. When the wells had dried, the spots were counted with an ELISpot reader (IRIS, Mabtech AB).

## mRNA preparation and quantification

RNA was extracted from cultured cells using the RNAmini kit (Thermo Fisher Scientific) and converted to cDNA (0.3 µg) as previously described [24]. The quantification of mRNA was performed using TaqMan real time PCR primer probes (Table 1, Applied Biosystems Thermo Fisher Scientific) on an Applied Biosystems 7500 Fast Real-Time PCR system (Thermo Fisher Scientific). To estimate relative expression, cycle counts were compared with the expression of housekeeping genes (PPIA and GAPDH). The mean CT values of the two genes were used to calculate the delta CT of each target gene.

## Statistics

All statistical analyses were conducted in Prism versions 9 and 10 by GraphPad Software. T-tests were used for normal distributions and ratio t-tests for log-normal distributions, as determined by Shapiro-Wilk normality and lognormality tests.

**Table 1.  TaqMan primers/probes targeting specific mRNAs used for mRNA quantification.**

| Gene | ID |
|------|-----|
| Peptidyl-prolyl cis-trans isomerase A (PPIA) | Hs99999901_g1 |
| Glyceraldehyde-3-phosphate dehydrogenase (GAPDH) | Hs02786624_g1 |
| Human TNF-α | Hs00176128_m1 |
| Human IL-1β | Hs01555410_m1 |
| Human IL-6 | Hs00174131_m1 |
| Human Foxp3 | Hs01085834_m1 |
| Human ISG20 (CD25) | Hs00158122_m1 |
| Human IFN-γ | Hs00980290_g1 |
| Human IL-10 | Hs00961622_m1 |
| HLAG | Hs00365950_g1 |
| HLAE | Hs03045171_m1 |

mRNA: messenger Ribonucleic Acid, TNF: tumor necrosis factor, IL: interleukin, CD: cluster of differentiation, IFN: interferon, HLA: human leukocyte antigen.

Simple linear regression was used for kinetic and dose-response analyses. Analysis of variance (ANOVA) with multiple comparisons was used in some analyses, both parametric and non-parametric (Friedman test), depending on normality.

## Results

### Comparison of culture media

In previous reports, cells were generated in ALyS505N-0 [17] or AIMV (Bashuda 2005) [16]. Here, we compared cells generated in TexMACS medium to those cultured in ALyS505N-0 or AIMV. Cell populations, cytokine concentration and production were examined, and no significant differences were noted between the three groups (S2–S5 Figs). TexMACS was also equivalent to ALyS505N-0 or AIMV in cell count, viability, and %Treg (S1 Table).

### Effect of supplementation with autologous plasma

Previous studies used incomplete media with 1% autologous plasma [16,17]. However, as we use use a complete medium, we assessed the need for supplementation of 1% autologous plasma in this setting. The addition of plasma to the TexMACS medium did not affect cell yields, viability, or cell composition (S6–S7 Figs). Culture supernatant from the group with added plasma contained significantly lower IL-10 (p = 0.0319) on day 14 compared to the group without plasma, but no significant differences were noted in the concentration of IL-6, IFN-γ, IL-1β, or TNF-α (Fig 1).

### Effect of RBC lysis

As leukapheresis products may be contaminated by a substantial number of remaining RBCs, we assessed the effect of RBC lysis on the generated product. There were no significant differences in DSIMC yield or viability (S8 Fig). The PBMC composition did not change significantly (% of T cells, B cells, and NK cells in total lymphocytes and % of Treg in CD4 + T cells, S9 Fig). Culture supernatant on day 7 from ACK-treated responder cells contained significantly lower concentrations of IL-6 (p = 0.0102), TNF-α (p = 0.049), IL-10 (p = 0.0458), and IL-1β (p = 0.0028) (Fig 2), suggesting a reduced non-specific inflammatory response.

The antigen-specific responses of the DSIMC were examined by cytokine ELISpot assay. DSIMC were restimulated with irradiated stimulator PBMC for 24 hours. IL-10 production was significantly higher (p = 0.0399) and IFN-γ lower

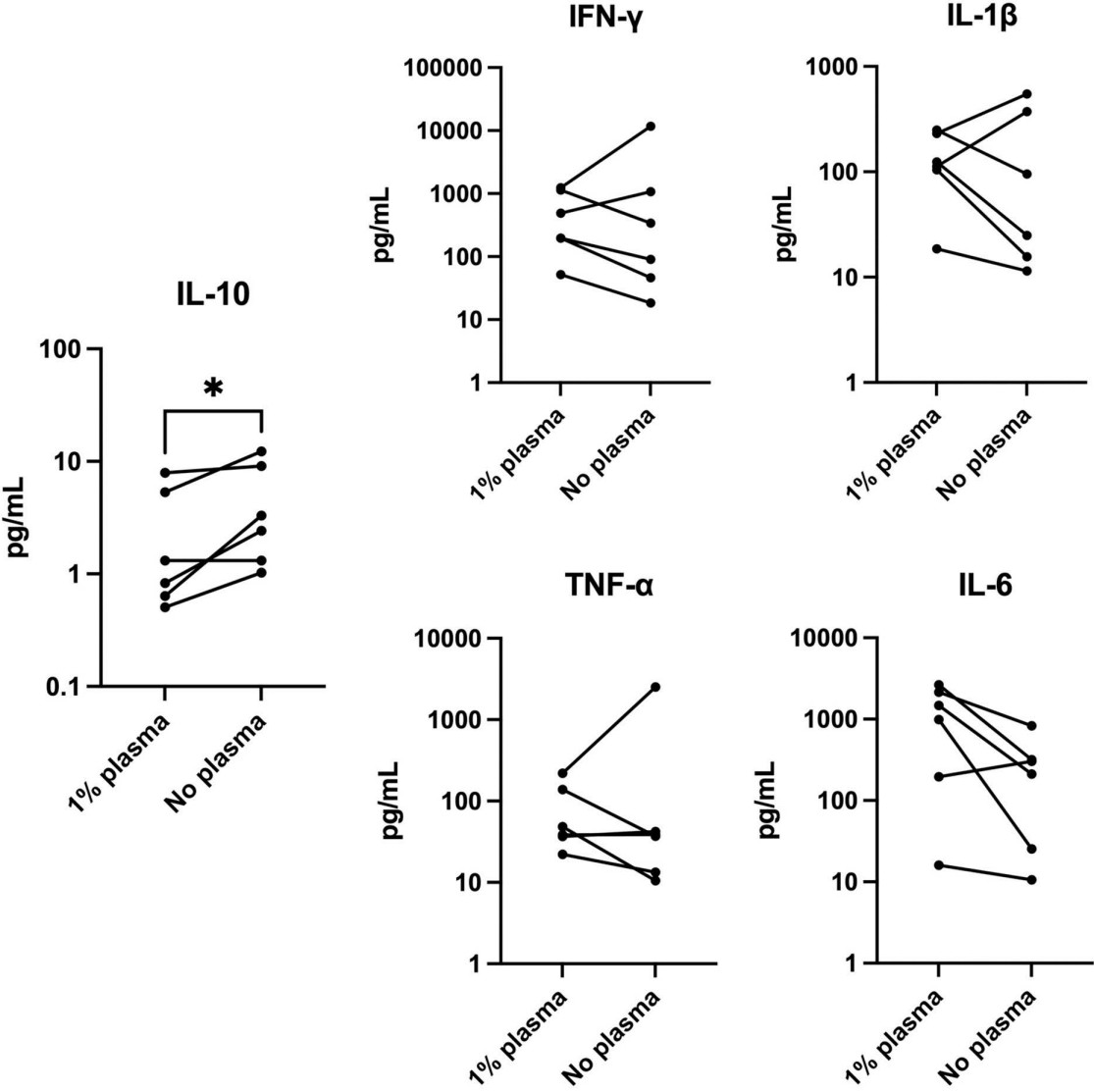

**Fig 1. Cytokine concentration in supernatant on culture day 14.** Comparison of PBMC co-cultures with and without 1% autologous recipient plasma. IL-10 was significantly lower with the addition of plasma. Differences in other measured cytokines did not reach significance. ECL. (Paired ratio t test, n = 6). * p < 0.05. PBMC: Peripheral blood mononuclear cells, IL: interleukin, ECL: electrochemiluminescence.

(p = 0.0055) in the ACK-treated group, indicating the treatment may improve specific antigen immunomodulatory function (Fig 3).

### DSIMC kinetics

To establish what changes were occurring in the DSIMC product over the second week of culture, we assessed the product after replenishment of stimulator PBMC on days 9, 12, and 14. Both responder and stimulator PBMC were treated with ACK, and DSIMC were generated in TexMACS medium. Cell number ($r^2 = 0.8516$, $p < 0.0001$) and viability ($r^2 = 0.8516$, $p < 0.0001$) decreased steadily over the two-week period (Figs 4a, S10). The NK cell population fraction of total lymphocytes was also reduced ($r^2 = 0.468$, $p = 0.0421$) (Fig 4b).

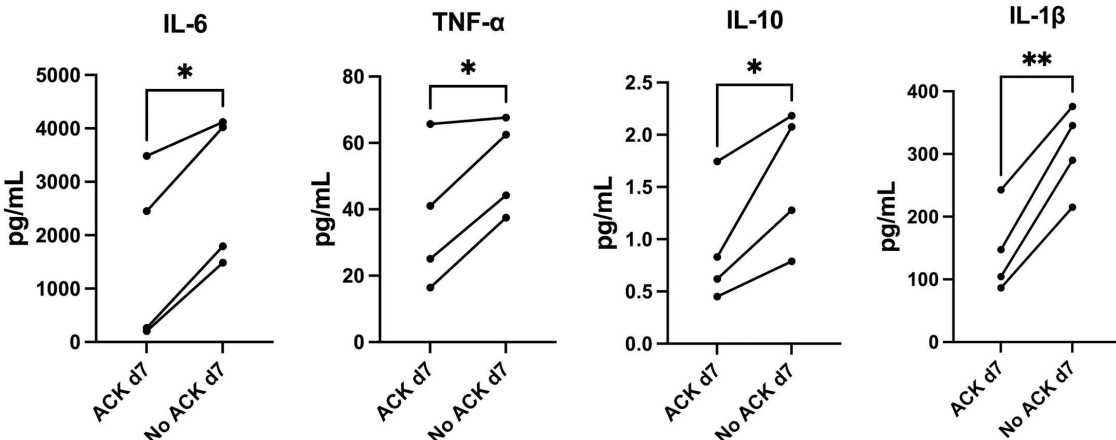

**Fig 2. Comparison of cytokine concentration in MLR supernatant after a 14-day co-culture with or without ACK treatment.** Cytokine concentration in culture supernatant (pg/mL) on day 7. IL-6, TNFα, IL-10, and IL-1β were all significantly lower in ACK-treated cultures on day 7. ECL. (Paired t-test, n = 4). * p < 0.05, ** p < 0.005. PBMC: Peripheral blood mononuclear cells, ACK: Ammonium Chloride Potassium lysis buffer, RBC: red blood cells, FCM: flow cytometry, IL: interleukin, TNF: tumor necrosis factor, ECL: electrochemiluminescence.

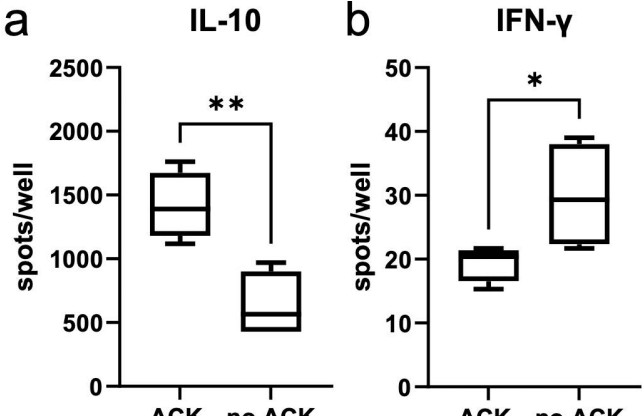

**Fig 3. Comparison of cytokine-production of stimulated DSIMC, with and without initial RBC lysis.** (a) IL-10 production 24 hours after restimulation was significantly higher, and (b) IFN-γ was significantly lower in the treated group. ELISpot. (Bars show means with IQR±SE, unpaired t test, n = 4). * p < 0.05, ** p < 0.01. DSIMC: donor specific immunomodulatory cells, ACK: Ammonium Chloride Potassium lysis buffer, RBC: red blood cells, CD: cluster of differentiation, IL: interleukin, IFN-γ: interferon gamma, ELISpot: Enzyme-Linked Immunosorbent spot.

Relative RNA gene expression of IFNG decreased (p = 0.0347) between days 12 and 14, while expression of IL10 increased (p = 0.0112) (Fig 5a,b). Similarly, ELISpot assays of cytokine production after specific stimulation showed a decrease in IFN-γ response between days 9 and 14 (p = 0.0211) (Fig 5c). There was a significant increase in IL-10 production between days 9 and 14 when broadly stimulated (p = 0.0447), but not in other groups (Fig 5d).

## Fate of stimulator cells

As remaining stimulator cells may be considered a contaminant in the generated product, the fate of these cells was investigated. Thawed irradiated HLA-A2− stimulator PBMC were added to co-cultures on day 7 with HLA-A2+ responder

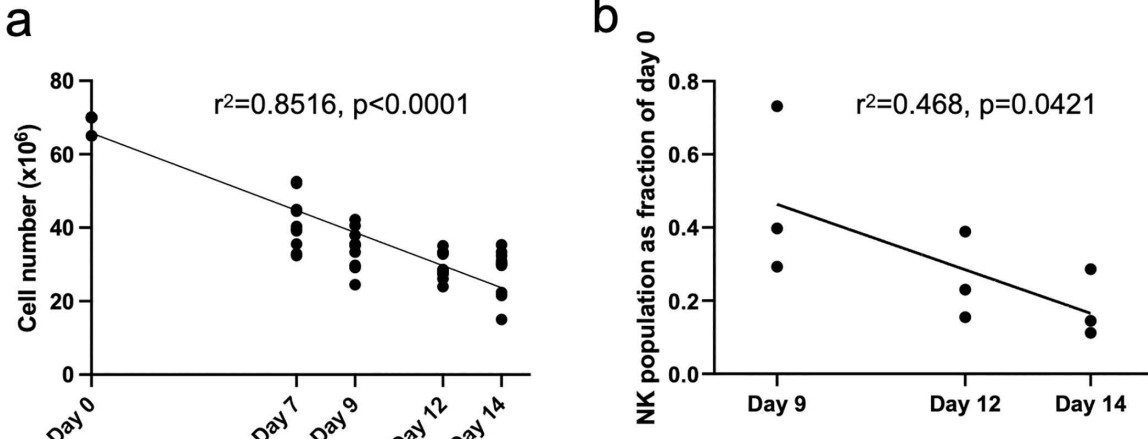

**Fig 4. Comparison of DSIMC generating co-cultures in the second week of incubation on days 9, 12 and 14.** (a) The total cell number reduces linearly over time (n = 10). Cell number estimated using automated cell counter. (b) The NK population (CD3-CD16 + CD56 + fraction of all CD45 +, here shown as a fraction of initial number on day 0) decreased linearly over time (n = 3). FCM. (Linear regression with extra-sum-of-squares F test). NK: natural killer cell, CD: cluster of differentiation, FCM: flow cytometry.

PBMC, and the fraction of HLA-A2+ cells was measured. DSIMCs cultured for 14 days contain a large amount of dead and apoptotic cells, which can easily be differentiated as they are small and highly granulated. Most irradiated stimulator cells had disappeared or were seen in the apoptotic population on day 14, while 96.5% of live lymphocytes were HLA-A2+, and therefore of responder origin (Fig 6). In the complementary experiment (HLA-A2+ irradiated stimulator, with HLA-A2− responder cells), 13.8% of live lymphocytes were HLA-A2+ two days after stimulator replenishment, which further decreased to 1.3% three days later (day 12) (S11 Fig).

### Belatacept titration

To establish an optimal concentration of belatacept, its effect on cytokine production and DSIMC composition in this culture protocol was evaluated. The concentrations tested were based on previous studies and were replicated to enable direct comparison. Davies et al. compared 0, 10 and 40 µg belatacept per µg/$10^6$ cells and found a maximal inhibition of the alloresponse at 40 µg, comparable to the use of 10 µg anti-B7 antibodies [22]. We extended the tested concentrations to include 0.1, 1 µg, and 80 µg. The percentage of Treg in CD4 + T cells showed a non-significant correlation with higher belatacept concentration (Fig 7a-b). There was a dose-dependent decline of PD-1 expression in Treg (FCM, $r^2 = 0.29$, p = 0.0067, Fig 7c), suggesting improved immunomodulatory function. In these experiments, IFN-γ concentration and production was completely suppressed at the lowest tested dose (0.1 µg/$10^6$ cells), while IL-10 did not significantly change (S12–S13 Figs). Cell numbers, viability, and fractions of T cells, B cells and NK cells in DSIMC were not correlated with dose (S14-S15 Figs). HELIOS expression in the Treg population was not affected (S16 Fig).

### Discussion

In this study, we aimed to optimize the in vitro production of donor-specific immunomodulatory cells (DSIMC). ACK lysis buffer-treated PBMC cultured in TexMACS medium supplemented with 1% responder plasma reliably generated DSIMC that meet established quality criteria. Further, we have determined a belatacept concentration of 40 µg/$10^6$ cells results in the survival of most recipient lymphocytes, with preserves functionality and acceptable stability. Most co-cultured stimulator cells are dead or apoptotic by day 12–14. Deviations from this suggested protocol did not improve measured outcomes.

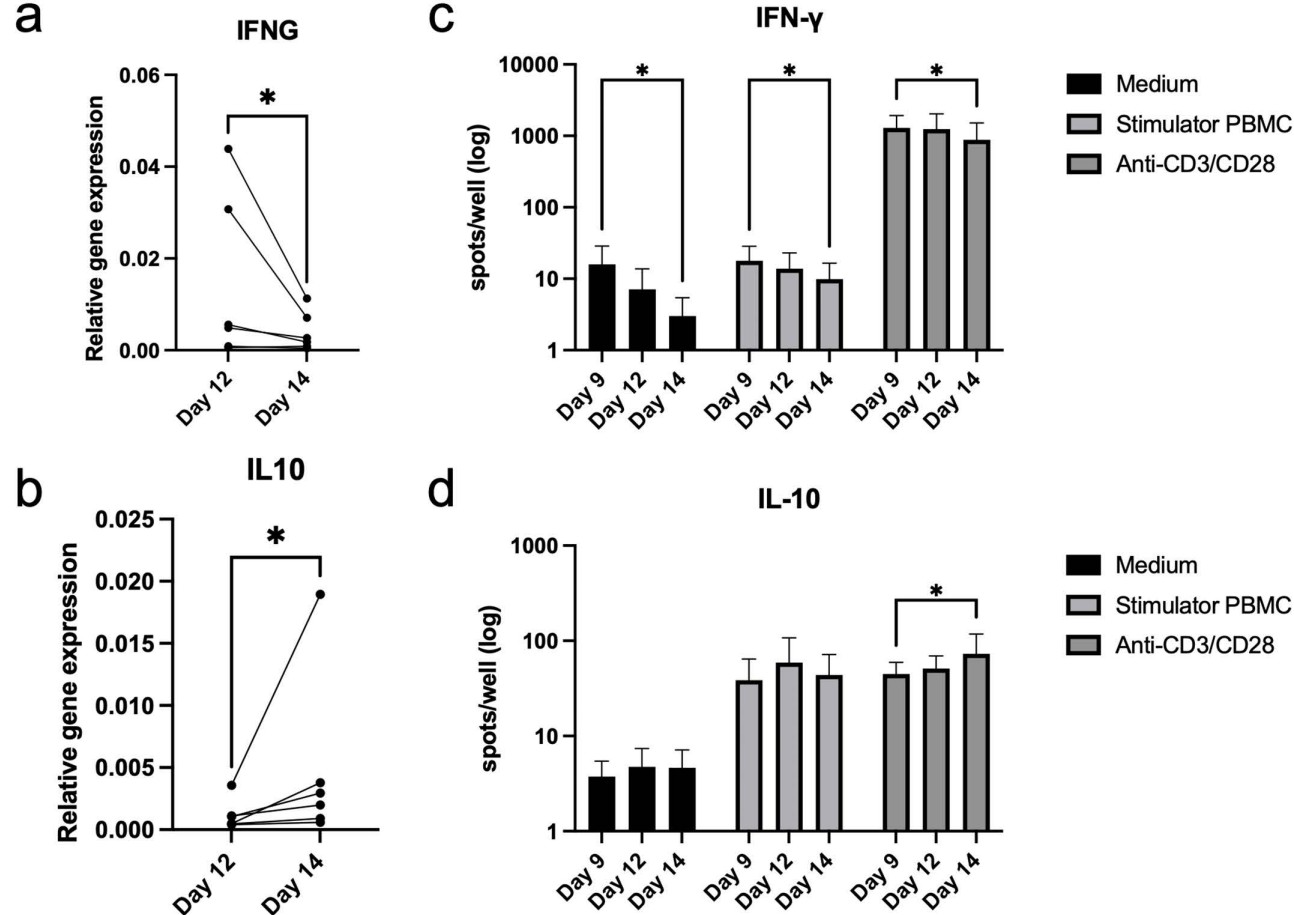

**Fig 5. Changes in cytokine production over the second week of culture.** (a,b) Higher IFNG and lower IL10 RNA gene expression on day 14 compared to day 12, relative to housekeeping gene. qPCR. (Paired t-test, n = 6). (c) IFN-γ production in both specific stimulator and negative control groups were significantly lower compared to the positive control (anti-CD3/CD28). The decrease in IFN-γ production between days 9 and 14 was significant in all groups. (d) IL-10 production was significantly increased in both specific stimulator and positive control groups compared to the negative control. ELISpot. 2-way ANOVA with multiple comparisons. n = 7 for IFN-γ, n = 4 for IL-10. Bars show means with SD, logarithmic y-axis. * p < 0.05, ** p < 0.01, *** p < 0.001, **** p < 0.0001. IFN: interferon, ELISpot: enzyme-linked immunosorbent spot, IL: interleukin, RNA: ribonucleic acid, qPCR: quantitative polymerase chain reaction, PBMC: peripheral blood mononuclear cell, CD: cluster of differentiation, ANOVA: analysis of variance.

TexMACS is a complete medium engineered to be used without additional human plasma or serum and was chosen for its suitability for T cell culture, GMP grade, and ease of accessibility. Bashuda et al.'s protocol consistently used ALyS505N-0 or AIM-V incomplete culture media supplemented with 1% autologous plasma [16,17], and there was remaining uncertainty as to the importance of this difference. We therefore examined the effect of DSIMC generation in TexMACS medium with and without 1% autologous plasma. Cell yield, viability, and cell composition did not significantly differ. However, significantly lower IL-10 concentration was noted in cultures with 1% plasma compared to those without plasma. This finding suggests a potential subtle influence of plasma components on the cytokine milieu within the culture. Interpreting these results is challenging due to the significant intra-experimental variance inherent in human samples, and the lack of actual donor and patient samples. The cytokine levels in the added plasma cannot solely account for the differences in concentrations observed in the culture supernatant, especially as it is sourced from healthy donors. Our clinical study looks at transplantation from BDD instead of LD, which was the focus of previous studies [17]. This difference could

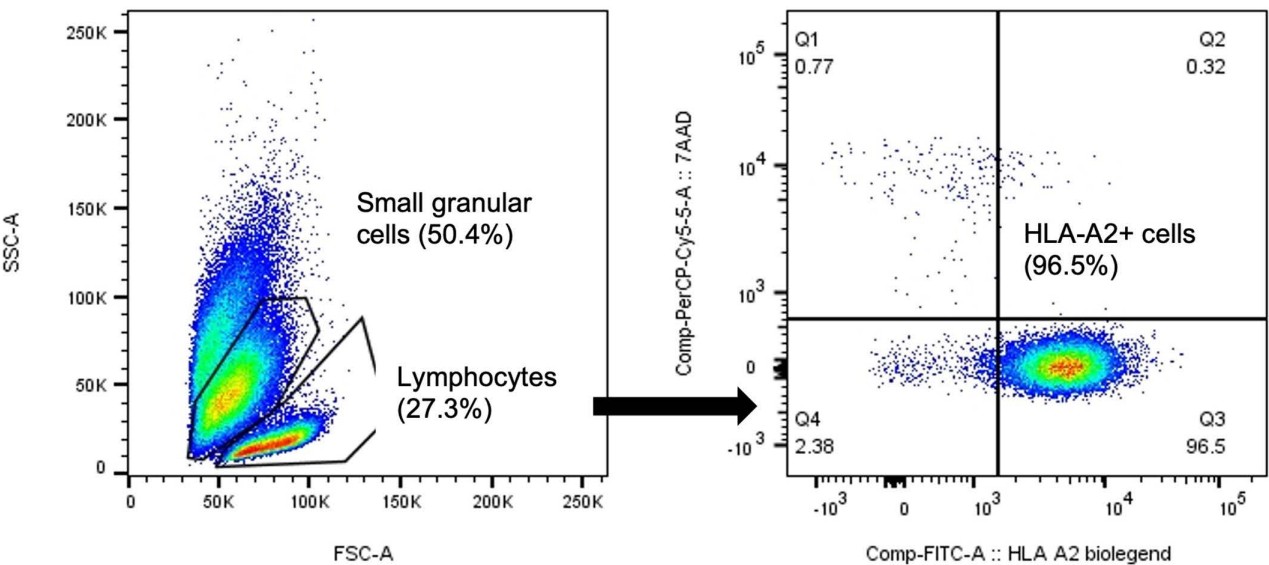

**Fig 6. Very few viable stimulator PBMC remain at end of culture.** HLA-A2+ responder PBMC co-cultured with HLA-A2− stimulator PBMC on culture day 14. Two experiments with mismatched HLA-A2 positive and negative responders and stimulators were conducted, showing most stimulator cells were dead or apoptotic by day 12-14. Co-cultured DSIMC were stained for HLA-A2 and 7AAD (dead cell marker). Most small granular cells were HLA-A2− and 7AAD+. FCM. Q3: HLA-A2+ responder lymphocytes. Q4: Irradiated HLA-A2− stimulator lymphocytes. HLA: human leukocyte antigen, PBMC: peripheral blood mononuclear cell, DSIMC: donor specific immunomodulatory cell, FCM: flow cytometry.

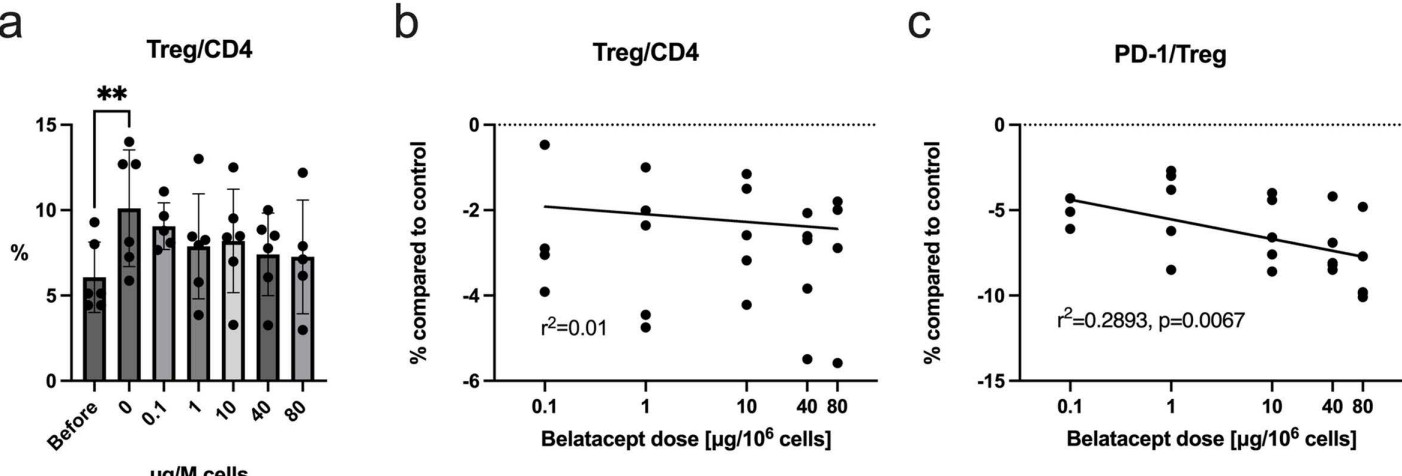

**Fig 7. Effect of different concentrations of belatacept on generation of DSIMC.** (a) Comparison of effect on Treg population before and after culture. Only the before/after comparison of the group without added belatace). (b) pt reached significance. FCM. Mixed-effects analysis with multiple comparisons. (n = 6The effect of concentration of belatacept on generation of DSIMC after two weeks. Treg percentage of CD4 + compared to same setting without added belatacept. Higher dose of belatacept was weakly but non-significantly correlated with a lower % of Treg, $r^2$ = 0.01. FCM. (Linear regression, n = 6). (c) Lower PD-1 expression on surface of regulatory T cell population with increasing concentration of belatacept. PD-1 is expressed on activated effector T cells and a higher dose of belatacept was correlated with lower expression. FCM. (Linear regression with extra-sum-of-squares F test, n = 5). DSIMC: donor-specific immunomodulatory cells, PD: programmed cell death protein, CD: cluster of differentiation, FCM: flow cytometry.

be a significant factor, as the inflammatory cytokine milieu changes significantly after brain death [25,26]. However, previous studies have demonstrated that inflammation is particularly challenging to manage in patients with autoimmune liver disease, to a greater extent than in patients undergoing liver transplantation for other indications [20]. Further investigation into the impact of recipient plasma on DSIMC generation is therefore warranted. While the full effect remains uncertain, the addition of plasma was not deemed to have a significantly negative effect.

When PBMC are collected through leukapheresis, there is often a substantial amount of contamination of RBC [27], and there can be up to 5 times more RBC than leukocytes in leukapheresis samples, with large operator and donor-dependent variability. This contamination affects cell counting and may also affect the development of generated cells. Therefore, ACK lysis buffer to lyse RBC was included in the standard operating procedure. ACK is routinely used in analysis. However, it has been shown that RBC lysing buffers can subtly change the composition of PBMC [28], and previous pilot studies performed in living donor transplantation projects did not utilize lysis buffers [16,20]. Therefore, we investigated its effect on DSIMC generation for this cell therapy application. In our *in vitro* experiments it reduced inflammatory activation, as indicated by lower proinflammatory cytokine production early in the culture period (Fig 2). Our experiments also indicated that ACK-treated PBMC develop a higher donor-specific immunomodulatory response as compared to cells generated without ACK treatment. This suggests that RBC lysis before culture may benefit the production of DSIMC.

The effect of different concentrations of belatacept on DSIMC generation has previously been examined by Davies et al. [22], upon which we had previously based our study comparing co-stimulation blocking reagents [23]. In the present study, we further titrated the dose down to include 0.1, 1 µg as well as doubling the dose to 80 µg, and compared cell numbers, viability, cell composition, and cytokine production. The effect on cytokine production was saturated far below the concentration that had previously been established as most efficacious for the inhibition of alloresponses by Davies et al. [22]. The percentage of regulatory T cells compared well with Teraoka et al.'s kidney study [19], but not with Todo et al.'s liver study [17]. The discrepancy in percentage of regulatory T cells between these studies could be explained by differences in FCM gating and staining techniques. Our previous study indicated there were no clear differences between the DSIMC generated with anti-CD80/CD86 antibodies and belatacept [23]. However, compared with direct inhibition of CD80/86 with specific antibodies, blocking the costimulatory complex with belatacept may have off-target effects, some of which may affect the normal function of regulatory T cells. CTLA-4 is expressed by regulatory T cells [13,29], and further addition may lead to competitive inhibition. We briefly looked at T cell activation and Treg-related markers, including PD-1 [30,31]. We saw a modest but significant decrease in the PD-1 expression of the Treg population with increasing concentration of belatacept (Fig 7). This effect was not seen in the CD3+T cell population. Previous research in mice has shown that a decrease in PD-1 in Treg cells may enhance their immunosuppressive function [32]. Previous studies using anti-CD80/86 did not look at these markers. It remains unclear whether this influences the generation of DSIMC *in vitro*, and its clinical significance is subject to even greater uncertainty. As part of the kinetics and titration studies, we also assessed the stability of the DSIMC Treg population. No significant differences in HELIOS expression were observed with varying belatacept concentrations or over time (S16 Fig). Similarly, FOXP3 gene expression (S17 Fig) and the production of IFN-γ and IL-10 (S18 Fig) remained stable over time. These findings suggest that the DSIMC Treg phenotype is maintained under the conditions tested. However, further investigation of the functional activity of Tregs within the DSIMC population is clearly indicated.

The effect of remaining irradiated stimulator cells being introduced to the transplant recipient is uncertain. While irradiated cells are unable to proliferate, they may be reactive and actively signal through cytokines and extracellular vesicles [33]. Therefore, we examined the number of viable stimulator cells in the generated product. Previous research had determined that most PBMC irradiated with as low as 6 Gy had disappeared after 6 days in culture [34]. To distinguish stimulator from responder cells in co-cultures, we stained PBMC with an anti-HLA-A2 probe, and cultured HLA-A2+ cells with HLA-A2− cells. The HLA serotype was selected for practical reasons, including the equal divide in relative frequency in the Swedish population [35]. By using this method, we confirmed that most irradiated stimulator cells disappear or are

apoptotic by day 14. Bashuda et al. and Todo et al. did not separate out apoptotic cells [16,17]. Removal of these cells from the final product would be costly in time and resources, and previous research with similar culture conditions and product compositions have shown promising results regardless. Considering the likely presence of apoptotic cells already being shed from the donated organ, the clinical significance of their removal from the DSIMC product may be limited. However, the potential impact of apoptotic cells on recipient immunity, including the release of DAMPs, remains of great interest, and further research is indicated. Suggested avenues of investigation include measuring donor DNA or mitochondrial DNA in plasma after transplantation or DSIMC injection.

The precise mechanisms of tolerance induction by DSIMC infusion remains to be fully elucidated. Potential contributing processes include direct suppression by DSIMC components, including regulatory T cells, and indirect effects on the recipient's immune environment. Further research will be crucial to elucidate their relative importance. Due to the success of previous clinical studies with living donors and promising preclinical results in primate models [16,20], this study focuses on optimizing the generation of DSIMC. Previous research including proliferation assays has shown promising DSIMC suppressive ability [23]. While further *in vitro* and *in vivo* studies are not feasible for this specific project, future research will explore more in-depth investigations, to more comprehensively evaluate this. Such assessments may include MLRs, Treg suppression assays, further assessment of cytokine profiles, FOXP3 TSDR demethylation assays, and single-cell RNA sequencing. Further enhancements to the method to optimize the production of DSIMC may also be warranted and may include other costimulation blockade agents and immunoregulatory cytokines. However, such use is currently complicated by the need for adhering to GMP standards. An assessment of the effect of metabolic states on tolerogenic differentiation may also yield interesting results. Preliminary feasibility batches, using blood from BDD and from patients on the list for liver transplantation, have already been performed in a GMP facility in preparation for clinical application [36]. Evaluations of sterility, endotoxin levels, scalability, cryopreservation viability, and batch-to-batch consistency indicate that generated DSIMC comply with the release criteria described in Todo et al's clinical trial [20] and with the standards set by the Swedish Medical Products Agency (Läkemedelsverket).

A limitation of research in this field is the general lack of reliable biomarkers for tolerance development [37–39]. The identification of such markers would enable early assessment of therapeutic efficacy, obviating the need to rely solely on clinical outcomes. Similarly, it has not been shown which qualities in the final DSIMC cell therapy predict success. In previous studies, DSIMC exhibited a suppressive effect on responder cell proliferation, decreased production of IFN-γ, and higher IL-10 production after stimulation with specific antigens. Cell composition studies have shown that generated cells have a significantly higher fraction of regulatory T cells in the CD4+T cell compartment, as compared to freshly isolated responder PBMC [23]. Because of this, work on finding new biomarkers will be of critical importance.

## Conclusion

We present a method of culturing cells that will efficiently generate donor-specific immunomodulatory cells. When made in a GMP setting and used as a cell therapy product, these may enable IS minimization and even complete cessation. The protocol is currently under evaluation in a clinical trial.

## Supporting information

**S1 Fig. Example of typical FCM gating.** CD4+T cell and Treg gating with activation marker PD-1.
(TIF)

**S2 Fig. Medium comparison: Phenotypes.** Comparison of DSIMC generation in three different media (AlyS505, AIM-V, TexMACS) compared to freshly separated PBMC. CD3+, CD19+, NK cells and Treg percentage of total lymphocytes. CD4+ and CD8+ cells of CD3+ cells. No significant differences. FCM. (n=3).
(TIF)

**S3 Fig. Medium comparison, cells and viability.** Comparison of DSIMC generation in three different media (AlyS505, AIM-V, TexMACS) over two weeks of culture. Cell numbers in million cells. Viability determined through live/dead staining (Trypan Blue) and presented as unstained live cells divided by stained dead cells. No significant differences. Manual cell counting. (n = 3).
(TIF)

**S4 Fig. Medium comparison: Cytokine concentration.** Comparison of DSIMC generation in three different media (AlyS505, AIM-V, TexMACS). IFNγ and IL-10 in ng/mL culture supernatant on day 14. No significant differences. ELISA. (n = 3).
(TIF)

**S5 Fig. Medium comparison: Cytokine production.** Comparison of DSIMC generation in three different media (AlyS505, AIM-V, TexMACS). Granzyme B, IFNγ, IL-17 and IL-10 production shown as total SFU (spot forming units) per million cells. Autologous cells added to DSIMC as negative control, donor stimulator cells added to DSIMC as specific proliferation activator, no cells added to controls. No significant differences. ELISPOT. (n = 3).
(TIF)

**S6 Fig. Plasma comparison: Phenotypes.** Comparison of CD4/CD8 ratio and Treg of CD4 + T cells in DSIMC generated with or without 1% autologous plasma added to the culture medium. No significant differences. FCM. (n = 9).
(TIF)

**S7 Fig. Plasma comparison: Cells and viability.** Comparison of DSIMC generation after two weeks of culture with or without 1% autologous plasma added to the culture medium. Cell numbers in million cells. No significant differences. Automated cell counter with nuclear staining and live/dead marker. (n = 6).
(TIF)

**S8 Fig. ACK comparison: Cells and viability.** Comparison of DSIMC generation after two weeks of culture with or without ACK-treated PBMC. Cell numbers in million cells. No significant differences. Automated cell counter with nuclear staining and live/dead marker. (n = 5).
(TIF)

**S9 Fig. ACK comparison: Phenotypes.** Comparison of DSIMC generation after two weeks of culture with or without ACK-treated PBMC. CD3 + T cells, regulatory T cells, NK cells, and B cells as percentage of total lymphocytes. No significant differences. FCM. (n = 4).
(TIF)

**S10 Fig. Kinetics: Viability and Treg percentage.** Comparison of viability and Treg percentage over time. (Left) Differences in viability stabilized over time. Automated cell counter with nuclear staining and live/dead marker, linear regression. (n = 10). (Right) No significant differences in percentage of Treg of total CD4 + T cells over time. FCM, paired t-test. (n = 3).
(TIF)

**S11 Fig. Fate of HLA-A2 + stimulator cells over the second week of culture, FCM gating.** In this culture, irradiated HLA-A2 + stimulator cells reduce from 13.8% to 1.43% of the total T cell compartment.
(TIF)

**S12 Fig. Titration of belatacept: Effect on concentration of IFNγ and IL10.** Effect of increasing concentration of belatacept in culture after two weeks on the concentration of cytokines in supernatant (ng/mL). No significant differences. Titration of belatacept as µg/million cells. ELISA. (n = 6).
(TIF)

**S13 Fig. Titration of belatacept: Effect on production of IFNγ and IL10.** Comparison of DSIMC generation with increasing concentration of belatacept on production of cytokines in SFU (spot forming units). (A/D) without stimulation, (B/E) with specific stimulation, (C/F) with general stimulation. No significant differences. ELISpot. (n = 3).
(TIF)

**S14 Fig. Titration of belatacept: Effect on cell count and viability.** Comparison of DSIMC generation with increasing concentration of belatacept on cell numbers and viability after 14 days of culture. No significant differences. Automated cell counter with nuclear staining and live/dead marker. (n = 6).
(TIF)

**S15 Fig. Titration of belatacept: Effect on cell phenotypes.** Comparison of DSIMC generation after two weeks of culture with increasing concentration of belatacept. T cells, NK cells, and B cells as percentage of total lymphocytes. No significant differences. FCM. (n = 6).
(TIF)

**S16 Fig. Treg stability: HELIOS expression.** (Left) Comparison of HELIOS expression in Treg population over time. No significant differences. ANOVA. Means±SD. FCM. (n = 5). (Right) Comparison of HELIOS expression in Treg population of DSIMC after two weeks of culture with increasing concentration of belatacept. No significant differences. Mixed-effects analysis. FCM. (n = 6, two missing datapoints).
(TIF)

**S17 Fig. Treg stability: PBMC FOXP3 expression.** Comparison of FOXP3 expression in Treg population over second week of culture. No significant differences. One-way ANOVA. qPCR. (n = 6).
(TIF)

**S18 Fig. Treg stability: Cytokine production.** Comparison of IFNγ and IL-10 concentration in culture supernatant over second week of culture. No significant differences. One-way ANOVA. Means±SEM. ELISA. (n = 6).
(TIF)

**S1 Table. Cell yield, viability and %Treg of generated DSIMC in three different media.** Treg (CD4＋CD25＋CD127lowFoxP3＋/CD45+) determined by FCM (n = 3).
(DOCX)

## Author contributions

**Conceptualization:** Nils Ågren, Ming Yao, Makiko Kumagai-Braesch.

**Data curation:** Nils Ågren, Makiko Kumagai-Braesch.

**Formal analysis:** Nils Ågren, Carl Skantze, Keyvan Habibi, Makiko Kumagai-Braesch.

**Funding acquisition:** Bo-Göran Ericzon, Makiko Kumagai-Braesch.

**Investigation:** Nils Ågren, Ming Yao, Carl Skantze, Keyvan Habibi, Makiko Kumagai-Braesch.

**Methodology:** Nils Ågren, Ming Yao, Makiko Kumagai-Braesch.

**Project administration:** Bo-Göran Ericzon, Makiko Kumagai-Braesch.

**Resources:** Makiko Kumagai-Braesch.

**Supervision:** Ming Yao, Bo-Göran Ericzon, Makiko Kumagai-Braesch.

**Validation:** Nils Ågren, Makiko Kumagai-Braesch.

**Visualization:** Nils Ågren.

**Writing – original draft:** Nils Ågren.

**Writing – review & editing:** Nils Ågren, Ming Yao, Carl Skantze, Keyvan Habibi, Bo-Göran Ericzon, Makiko Kumagai-Braesch.

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
