## [Decision Letter · Decision Letter 0]

15 Jul 2025

Dear Dr. Ågren,

Thank you for submitting your manuscript to PLOS ONE. After careful consideration, we feel that it has merit but does not fully meet PLOS ONE’s publication criteria as it currently stands. Therefore, we invite you to submit a revised version of the manuscript that addresses the points raised during the review process.

Please respond to reviewers' comments individually.

We look forward to receiving your revised manuscript.

Kind regards,

Xiaosheng Tan

Academic Editor

PLOS ONE

Journal Requirements:

[This study was supported in part by research grants from the Swedish Research Council (Vetenskapsrådet, VR2018-00845), the regional agreement on medical specialization training and clinical research between the Stockholm County Council and Karolinska University Hospital (20180606), CIMED (Center for Innovative Medicine, FoUI-975210), and Gelinfonden. Bristol-Myers Squibb (BMS) provided belatacept for experimental use but had no role in the study design, data collection, analysis, or manuscript preparation.].

Reviewers' comments:

Reviewer's Responses to Questions

**Comments to the Author**

1. Is the manuscript technically sound, and do the data support the conclusions?

Reviewer #1: Partly

Reviewer #2: Yes

Reviewer #3: Yes

2. Has the statistical analysis been performed appropriately and rigorously?

Reviewer #1: Yes

Reviewer #2: Yes

Reviewer #3: Yes

3. Have the authors made all data underlying the findings in their manuscript fully available?

Reviewer #1: Yes

Reviewer #2: Yes

Reviewer #3: Yes

4. Is the manuscript presented in an intelligible fashion and written in standard English?

Reviewer #1: Yes

Reviewer #2: Yes

Reviewer #3: Yes

Reviewer #1: This small study by Nils Ågren et al. presents a protocol for optimizing DSIMC production by treating responder and irradiated stimulator PBMCs with RBC lysis buffer and culturing them in TexMACS medium supplemented with 1% autologous plasma and 40 μg per million cells of belatacept for 14 days. The study has been performed well and the manuscript is clearly written, although there are several issues that should be addressed as listed below.

1. The sentence: “In a previous study we used cells from healthy blood donors to simulate the co-culture of organ donor and recipient cells as stimulators and responders, comparing the effects of belatacept and mouse anti-human CD80/CD86 antibodies, and found them to be equal [17].” This reference seems to be incorrect here.

2. The research on Belatacept titration should be more systematic to provide readers with clearer information, including more appropriate concentration levels.

3. CTLA-4 is universally present on FOXP3+ Tregs. Since CD28 signaling appears essential to generating and maintaining CD4+CD25+ Tregs and co-stimulation blockade induces T-cell anergy and apoptosis, CTLA-4 Ig therapy, therefore, might negatively affect Treg generation. The author should provide more research results on DSIMC functions to facilitate a more in-depth discussion of the pros and cons of using belatacept to induce DSIMC production.

4. Page 7, lines 136-138: details quantity of the thawed stimulator cells should be provided.

5. In S10 Fig, it would be better to provide the percentage change of Treg cells within the total CD4+ T cells from Day 0 to Day 14.

6. Given the significance of Belatacept in this study, I suggest that S12 Fig be placed in the main text figures.

Reviewer #2: This study investigates the optimal culture conditions for DSIMC production, including comparison of culture media, autologous plasma supplementation, the effect of RBC lysis, DSIMC kinetics, the fate of stimulator cells, and belatacept titration. The authors ultimately propose a protocol involving treatment of both responder and irradiated stimulator PBMCs with RBC lysis buffer, followed by culture in TexMACS medium supplemented with 1% autologous plasma and 40 μg per million cells of belatacept for 14 days, to optimize DSIMC generation.

The manuscript presents a potentially clinically impactful protocol for generating donor-specific immunomodulatory cells. However, the major concern lies in the structure and clarity of the manuscript. Some sections of the Methods are written more like descriptive results. Similarly, the Results are presented as a series of observations without sufficient rationale at the beginning of each subsection or a clear conclusion at the end. The authors are strongly encouraged to revise these sections to enhance the logical flow, scientific rigor, and overall readability of the manuscript.

Below are my specific comments, which I hope will assist in improving the quality of the manuscript:

1) Methods section: Plasma Preparation. The centrifugation speed is currently written as “2500 xg”, which is not the correct format.

2) Methods section: Effect of Addition of Autologous Plasma. It reads more like a rationale than a methodological description. Specifically, it includes background information better suited to the Results or Discussion. It lacks essential experimental details.

3) Methods section: Statistics. The acronym “ANOVA” should be written out in full when first mentioned in this manuscript.

4) Figure 1. Only IL-10 shows a sample size of n =6, while the other cytokines appear to have fewer replicates. Since it seems that the samples are derived from the same groups and only the cytokines differ, please clarify why the number of measurements varies among cytokines.

5) Figure 2. The figure is missing subfigure labels (a, b, c, d), which are referenced in the figure legend. Additionally, the order of cytokines in the legend (“IL-6, IL-10, IL-1β, and TNFα”) does not match their actual order in the figure (“IL-6, TNFα, IL-10, IL-1β”).

6) Figure 3. The figure legend mentions the term “PBMC”, but no such term appears in this part.

7) Line 329. A comma is missing before “CD”.

8) Results. Each section of the Results should begin with a brief rationale to provide context and explain the purpose of the experiments. Additionally, each section should conclude with a clear summary of the findings.

Reviewer #3: Nils et al. investigated the optimization of in vitro production of immunomodulatory cells (DSIMCs) for the induction of donor-specific tolerance in solid organ transplantation. While the study offers valuable preliminary insights into the generation and function of DSIMCs, including the impact of belatacept concentration, red blood cell lysis, and plasma supplementation, several areas require further experimental validation and methodological refinement to strengthen the conclusions and enhance the translational potential of this cell therapy approach.

Major Comments:

1. A deeper mechanistic understanding of the regulatory function of DSIMCs is essential. Functional suppression assays using labeled responder T cells should be employed to assess the suppressive capacity of induced Tregs. Additionally, mixed lymphocyte reactions using third-party PBMCs can confirm the donor-specific nature of the induced tolerance. Evaluating the stability of the regulatory phenotype through FOXP3 TSDR demethylation and Helios expression, combined with longitudinal restimulation assays and cytokine profiling (e.g., IL-10, IFN-γ), would offer more robust evidence for the durability and specificity of DSIMCs.

2. The phenotypic and functional heterogeneity of DSIMCs should be further investigated using high-dimensional techniques such as flow cytometry. This would allow precise identification of immunomodulatory subsets including classical Tregs, Tr1 cells, tolerogenic dendritic cells, and myeloid-derived suppressor cells. Additionally, single-cell RNA sequencing or CITE-seq could uncover the transcriptional programs and cell states that underlie the tolerogenic potential of the DSIMC population.

3. While belatacept concentration and plasma supplementation were examined, further optimization of the culture environment is warranted. This includes testing the effects of low-dose immunoregulatory cytokines (e.g., IL-2, TGF-β, IL-27), culturing under physiologic hypoxia, and assessing metabolic programming using Seahorse assays to determine whether tolerogenic differentiation is associated with distinct metabolic states. The incorporation of alternative costimulatory blockade agents, such as anti-CD40L or JAK inhibitors, may further enhance the immunomodulatory properties of DSIMCs.

4. To move toward clinical application, the DSIMC production protocol should be adapted to GMP standards. This includes evaluation of sterility, endotoxin levels, scalability, cryopreservation viability, and batch-to-batch consistency.

**Do you want your identity to be public for this peer review?** For information about this choice, including consent withdrawal, please see our Privacy Policy

Reviewer #1: No

Reviewer #2: No

Reviewer #3: No

---

## [Author Response · Author response to Decision Letter 1]

1 Sep 2025

Reviewer #1:

This small study by Nils Ågren et al. presents a protocol for optimizing DSIMC production by treating responder and irradiated stimulator PBMCs with RBC lysis buffer and culturing them in TexMACS medium supplemented with 1% autologous plasma and 40 μg per million cells of belatacept for 14 days. The study has been performed well and the manuscript is clearly written, although there are several issues that should be addressed as listed below.

1. The sentence: “In a previous study we used cells from healthy blood donors to simulate the co-culture of organ donor and recipient cells as stimulators and responders, comparing the effects of belatacept and mouse anti-human CD80/CD86 antibodies, and found them to be equal [17].” This reference seems to be incorrect here.

Response: We thank the reviewer for pointing this out. We have corrected the reference. The reference number was changed to 23 (Watanabe et al., 2018) instead of 17 on line 105 in the manuscript with tracked changes.

2. The research on Belatacept titration should be more systematic to provide readers with clearer information, including more appropriate concentration levels.

Response: The concentrations chosen were based on previous research by Davies et al. and were replicated for the sake of comparison. Davies et al. compared 0, 10 and 40 μg belatacept per million cells and found a maximal inhibition of the alloresponse at 40 μg, comparable to the use of 10 μg anti-B7 antibodies. In this paper concentrations differing by several orders of magnitude were tested and included concentrations of 0.1 and 1 μg as well as a doubled concentration of 80 µg. While we agree these concentrations may be considered arbitrary on its face, we still believe the linear extrapolation to be valid (e.g. Fig 7). We expanded the description of the belatacept titration experiment in the Methods section with this more comprehensive rationale for the concentration range tested. Please see the Belatacept titration section in the Results (lines 435-447 in the manuscript with tracked changes).

3. CTLA-4 is universally present on FOXP3+ Tregs. Since CD28 signaling appears essential to generating and maintaining CD4+CD25+ Tregs and co-stimulation blockade induces T-cell anergy and apoptosis, CTLA-4 Ig therapy, therefore, might negatively affect Treg generation. The author should provide more research results on DSIMC functions to facilitate a more in-depth discussion of the pros and cons of using belatacept to induce DSIMC production.

Response: We acknowledge the complexity of CTLA-4 Ig effects and have elaborated on this in the Discussion section. Additionally, we included supplementary data characterizing Treg dynamics and functional markers in response to belatacept (HELIOS marker stability and FOXP3 expression). See the second half of the 4th paragraph in the Discussion (lines 551-579 in the manuscript with tracked changes).

4. Page 7, lines 136-138: details quantity of the thawed stimulator cells should be provided.

Response: The exact number of thawed stimulator cells (20 × 106) has been added. See Cell culture conditions in the Methods section (line 183 in the manuscript with tracked changes).

5. Improve S10 Fig by showing percentage change of Tregs. In S10 Fig, it would be better to provide the percentage change of Treg cells within the total CD4+ T cells from Day 0 to Day 14.

Response: This figure is meant to visualize the stability over the second week. For comparisons with and without belatacept from day 0, see reply to next comment and new Figure 7.

6. Given the significance of Belatacept in this study, I suggest that S12 Fig be placed in the main text figures.

Response: We agree and have removed S12 Fig and replaced it with a new Figure 7 which now contains a comparison of Treg% over time with different concentrations of belatacept.

Reviewer #2:

This study investigates the optimal culture conditions for DSIMC production, including comparison of culture media, autologous plasma supplementation, the effect of RBC lysis, DSIMC kinetics, the fate of stimulator cells, and belatacept titration. The authors ultimately propose a protocol involving treatment of both responder and irradiated stimulator PBMCs with RBC lysis buffer, followed by culture in TexMACS medium supplemented with 1% autologous plasma and 40 μg per million cells of belatacept for 14 days, to optimize DSIMC generation.

The manuscript presents a potentially clinically impactful protocol for generating donor-specific immunomodulatory cells. However, the major concern lies in the structure and clarity of the manuscript. Some sections of the Methods are written more like descriptive results. Similarly, the Results are presented as a series of observations without sufficient rationale at the beginning of each subsection or a clear conclusion at the end. The authors are strongly encouraged to revise these sections to enhance the logical flow, scientific rigor, and overall readability of the manuscript.

Below are my specific comments, which I hope will assist in improving the quality of the manuscript:

1. Methods section: Plasma Preparation. The centrifugation speed is currently written as “2500 xg”, which is not the correct format.

Response: This has been corrected to the standard “2500 × g”. See Plasma preparation in the Methods section (lines 157-160 in the manuscript with tracked changes).

2. Methods section: Effect of Addition of Autologous Plasma. It reads more like a rationale than a methodological description. Specifically, it includes background information better suited to the Results or Discussion. It lacks essential experimental details.

Response: We revised this section to focus strictly on methodology and moved rationale statements to the Discussion. See Effect of addition of autologous plasma in the Methods section (lines 201-204 in the Methods section in the manuscript with tracked changes).

3. Methods section: Statistics. The acronym “ANOVA” should be written out in full when first mentioned in this manuscript.

Response: “ANOVA” has been written out in full as “analysis of variance” on first use. See Statistics in the Methods section (line 307 in the manuscript with tracked changes).

4. Figure 1. Only IL-10 shows a sample size of n =6, while the other cytokines appear to have fewer replicates. Since it seems that the samples are derived from the same groups and only the cytokines differ, please clarify why the number of measurements varies among cytokines.

Response: All samples are n=6, as written. Visual overlap may explain the apparent lack of data points.

5. Figure 2. The figure is missing subfigure labels (a, b, c, d), which are referenced in the figure legend. Additionally, the order of cytokines in the legend (“IL-6, IL-10, IL-1β, and TNFα”) does not match their actual order in the figure (“IL-6, TNFα, IL-10, IL-1β”).

Response: References to subfigures (a–d) have been removed in the Figure 2 legend, and the legend now matches the order of cytokines displayed in the text. See Effect of RBC lysis in the Results section and the legend of Figure 2 (lines 345-346 and 348-350 in the manuscript with tracked changes).

6. Figure 3. The figure legend mentions the term “PBMC”, but no such term appears in this part.

Response: The term “PBMC” has been removed from the legend where it was not relevant. See Figure 3 legend (line 365 in the manuscript with tracked changes).

7. Line 329. A comma is missing before “CD”.

Response: Corrected as suggested.

8. Results. Each section of the Results should begin with a brief rationale to provide context and explain the purpose of the experiments. Additionally, each section should conclude with a clear summary of the findings.

Response: Each Results subsection now begins with a brief rationale.

Reviewer #3:

Nils et al. investigated the optimization of in vitro production of immunomodulatory cells (DSIMCs) for the induction of donor-specific tolerance in solid organ transplantation. While the study offers valuable preliminary insights into the generation and function of DSIMCs, including the impact of belatacept concentration, red blood cell lysis, and plasma supplementation, several areas require further experimental validation and methodological refinement to strengthen the conclusions and enhance the translational potential of this cell therapy approach.

Major Comments:

1. A deeper mechanistic understanding of the regulatory function of DSIMCs is essential. Functional suppression assays using labeled responder T cells should be employed to assess the suppressive capacity of induced Tregs. Additionally, mixed lymphocyte reactions using third-party PBMCs can confirm the donor-specific nature of the induced tolerance. Evaluating the stability of the regulatory phenotype through FOXP3 TSDR demethylation and Helios expression, combined with longitudinal restimulation assays and cytokine profiling (e.g., IL-10, IFN-γ), would offer more robust evidence for the durability and specificity of DSIMCs.

Response: Thank you for your valuable comments. We agree on the importance of functional validation. We did indeed perform a series of experiments using HELIOS antibodies as part of this series of experiments, which did not differ significantly over time, or with belatacept concentration. As suggested, this may speak to Treg stability. We had also performed a FOXP3 gene expression analysis using PCR and did not see any significant changes over the second week of culture. Cytokine concentration in supernatant, as assayed using ELISA (both IL-10 and IFN-γ), did not show significant changes over the second week. We have added these results as supporting information (S16-18) and these findings are now discussed in the Results and Discussion sections. Mixed lymphocyte reaction inhibition assays using third-party PBMCs were used in a previous study comparing belatacept to anti-CD80/86 monoclonal antibodies, which did indeed confirm donor-specificity (Watanabe et al 2018, see manuscript ref 23). While it would be highly interesting, we do not have the current ability to follow up these findings with suppression assays using CFSE-labeled responder cells, or FOXP3 TSDR analysis. This has also been acknowledged in the Discussion. See the 4th and 6th paragraphs in the Discussion section (lines 573-579 and 615-635 in the manuscript with tracked changes).

2. The phenotypic and functional heterogeneity of DSIMCs should be further investigated using high-dimensional techniques such as flow cytometry. This would allow precise identification of immunomodulatory subsets including classical Tregs, Tr1 cells, tolerogenic dendritic cells, and myeloid-derived suppressor cells. Additionally, single-cell RNA sequencing or CITE-seq could uncover the transcriptional programs and cell states that underlie the tolerogenic potential of the DSIMC population.

Response: We are currently trying to investigate the precise mechanisms. Myeloid-derived cells have so far not been a focus, but we are actively looking into them as part of a separate project. Plans for future single-cell RNA sequencing have been discussed and will also be part of future investigations. These suggestions are now discussed in the Discussion section. See the 6th paragraph in the Discussion section (line 621-623 in the manuscript with tracked changes).

3. While belatacept concentration and plasma supplementation were examined, further optimization of the culture environment is warranted. This includes testing the effects of low-dose immunoregulatory cytokines (e.g., IL-2, TGF-β, IL-27), culturing under physiologic hypoxia, and assessing metabolic programming using Seahorse assays to determine whether tolerogenic differentiation is associated with distinct metabolic states. The incorporation of alternative costimulatory blockade agents, such as anti-CD40L or JAK inhibitors, may further enhance the immunomodulatory properties of DSIMCs.

Response: We appreciate these suggestions and have included a discussion of potential enhancements. It will certainly be considered in future studies to evaluate generated cells. While such enhancements would be challenging to include in current clinical trials, it will be important to evaluate the possibility of further improving on the development of DSIMC going forward. See the 6th paragraph in the Discussion section (lines 623-628 in the manuscript with tracked changes).

4. To move toward clinical application, the DSIMC production protocol should be adapted to GMP standards. This includes evaluation of sterility, endotoxin levels, scalability, cryopreservation viability, and batch-to-batch consistency.

Response: In collaboration with the Vecura GMP lab at Huddinge, this has already been concluded. However, as this work does not fall under the remit of the manuscript outline, it has not been included. There is a short roadmap in the discussion that superficially covers this. See the 6th paragraph in the Discussion section (line 629-635 in the manuscript with tracked changes). For more information, see the entry in EMA:s Clinical Trials Information System at [https://euclinicaltrials.eu/search-for-clinical-trials/?lang=en&EUCT=2024-514866-37-01].

---

## [Decision Letter · Decision Letter 1]

14 Sep 2025

Optimizing the In Vitro Production of Immunomodulatory Cells for the Induction of Tolerance in Solid Organ Transplantation

PONE-D-25-19100R1

Dear Dr. Ågren,

We’re pleased to inform you that your manuscript has been judged scientifically suitable for publication and will be formally accepted for publication once it meets all outstanding technical requirements.

Kind regards,

Xiaosheng Tan

Academic Editor

PLOS ONE

Additional Editor Comments (optional):

Reviewer #1:

Reviewer #2:

Reviewer #3:

Reviewers' comments:

Reviewer's Responses to Questions

**Comments to the Author**

Reviewer #1: All comments have been addressed

Reviewer #2: (No Response)

Reviewer #3: All comments have been addressed

2. Is the manuscript technically sound, and do the data support the conclusions?

Reviewer #1: Yes

Reviewer #2: Yes

Reviewer #3: Yes

3. Has the statistical analysis been performed appropriately and rigorously?

Reviewer #1: Yes

Reviewer #2: Yes

Reviewer #3: Yes

4. Have the authors made all data underlying the findings in their manuscript fully available?

Reviewer #1: Yes

Reviewer #2: Yes

Reviewer #3: Yes

5. Is the manuscript presented in an intelligible fashion and written in standard English?

Reviewer #1: Yes

Reviewer #2: Yes

Reviewer #3: Yes

Reviewer #1: 作者对这六个问题进行了全面解决并进行了严格的修改。这些修订不仅在科学上合理和详细而且还满足了我的基本要求——例如通过将补充图 12 �S12� 集成到信息量更大的图 7 中。此外使用特定的跟踪更改显着提高了修改的透明度和可验证性。

Reviewer #2: (No Response)

Reviewer #3: Although not all of my previous suggestions were fully addressed, the authors have satisfactorily responded to my major concerns. In my opinion, the current version has been improved substantially and is suitable for acceptance.

**Do you want your identity to be public for this peer review?** For information about this choice, including consent withdrawal, please see our Privacy Policy

Reviewer #1: No

Reviewer #2: No

Reviewer #3: **Yes: ** Jing Ju

---

## [Editor Report · Acceptance letter]

PONE-D-25-19100R1

PLOS ONE

Dear Dr. Ågren,

I'm pleased to inform you that your manuscript has been deemed suitable for publication in PLOS ONE. Congratulations! Your manuscript is now being handed over to our production team.

Kind regards,

on behalf of

Dr. Xiaosheng Tan

Academic Editor

PLOS ONE